# Lamin Mutations Cause Increased YAP Nuclear Entry in Muscle Stem Cells

**DOI:** 10.3390/cells9040816

**Published:** 2020-03-27

**Authors:** Daniel J. Owens, Martina Fischer, Saline Jabre, Sophie Moog, Kamel Mamchaoui, Gillian Butler-Browne, Catherine Coirault

**Affiliations:** 1INSERM UMRS_974, Centre for Research in Myology, Sorbonne Université, 75013 Paris, France; D.J.Owens@ljmu.ac.uk (D.J.O.); mfischer.p@googlemail.com (M.F.); saline.j.jabr@net.usek.edu.lb (S.J.); gillian.butler-browne@upmc.fr (G.B.-B.); 2Research Institute for Sport and Exercise Science, Liverpool John Moores University, Liverpool L3 3AF, UK; 3Inovarion, 75013 Paris, France; sophie.moog@inovarion.com; 4Association Institut de Myology, 75013 Paris, France; kamel.mamchaoui@upmc.fr

**Keywords:** lamins, congenital myopathy, nucleo-cytoskeletal translocation, nuclear envelope

## Abstract

Mutations in the *LMNA* gene, encoding the nuclear envelope A-type lamins, are responsible for muscular dystrophies, the most severe form being the *LMNA*-related congenital muscular dystrophy (L-CMD), with severe defects in myonucleus integrity. We previously reported that L-CMD mutations compromise the ability of muscle stem cells to modulate the yes-associated protein (YAP), a pivotal factor in mechanotransduction and myogenesis. Here, we investigated the intrinsic mechanisms by which lamins influence YAP subcellular distribution, by analyzing different conditions affecting the balance between nuclear import and export of YAP. In contrast to wild type (WT) cells, *LMNA*^DK32^ mutations failed to exclude YAP from the nucleus and to inactivate its transcriptional activity at high cell density, despite activation of the Hippo pathway. Inhibiting nuclear pore import abolished YAP nuclear accumulation in confluent mutant cells, thus showing persistent nuclear import of YAP at cell confluence. YAP deregulation was also present in congenital myopathy related to nesprin-1 ^KASH^ mutation, but not in cells expressing the *LMNA*^H222P^ mutation, the adult form of lamin-related muscle dystrophy with reduced nuclear deformability. In conclusion, our data showed that L-CMD mutations increased YAP nuclear localization via an increased nuclear import and implicated YAP as a pathogenic contributor in muscle dystrophies caused by nuclear envelop defects.

## 1. Introduction

The nuclear lamins A/C are type V intermediate filament proteins encoded by the *LMNA* gene. Lamins form complexes with other proteins of the nuclear membrane to influence mechanical cues and signaling pathways crucial for cellular proliferation and differentiation [1]. Mutations in the *LMNA* gene cause laminopathies, a highly heterogeneous group of disorders, including muscular dystrophies and cardiomyopathies [2,3]. The disease mechanisms underlying *LMNA*-related muscular dystrophy remains somewhat elusive.

There is clear evidence that A-type lamins and nuclear envelope proteins play a critical role in responding to mechanical cues from the extracellular matrix by adjusting the cytoskeleton and nuclear stiffness with the stiffness of the tissue microenvironments [4,5]. Structural changes in lamin A/C can also affect several signaling pathways, by altering either direct or indirect interactions of signaling molecules with A-type lamins [6,7]. Lamins A/C have already been shown to regulate the nuclear translocation and downstream signaling of the mechanosensitive transcription factor megakaryoblastic leukemia 1 (MKL1), a myocardin family member pivotal in cardiac development and function [8]. In muscle stem cells (MuSCs) from patients carrying A-type lamin mutations, we recently reported an impaired ability to sense matrix stiffness [9] and to withstand mechanical stretching of the extracellular matrix, causing aberrant regulation of the yes-associated protein (YAP) [10].

YAP is a transcriptional co-regulator that is modulated by diverse biomechanical signals and transduces them into cell-specific transcriptional responses, regulating cell proliferation and survival, organ growth, stem-cell renewal, and cell differentiation [11]. A major mechanism of YAP regulation occurs at the level of its subcellular localization, as YAP nuclear accumulation promotes target gene transcription and cell proliferation (reviewed in [11,12]). After phosphorylation by LATS1/2 kinase, YAP binds to 14–3–3 proteins, leading to its cytoplasmic retention and degradation (reviewed in [11,12]), and favoring skeletal muscle differentiation [12].

A-type lamins influence the localization and transcriptional activity of YAP [10]. It has been shown that lamin-A overexpression decreases both total YAP levels and nuclear localization in mesenchymal stem cells [5]. In contrast, increased YAP nuclear localization and activity in combination with reduced lamin levels is observed in cancers of many organ types (reviewed in [13]), as well as in *LMNA* mutant MuSCs cultured on soft matrices [10]. However, it remains unclear as to how mutant lamins cause defects in the YAP signaling pathway. Abnormal nuclear shape is observed in diseases where the A-type lamins are altered, including cancer [14,15] and laminopathies [16]. Altered nuclear morphology can in turn increase the rate of YAP import [17,18], by opening up nuclear pores [17]. One can hypothesize that A-type lamin mutations, responsible for severe skeletal muscle laminopathies, will cause an increase YAP nuclear localization because of an increased nuclear import.

To test this hypothesis, we investigated YAP subcellular distribution/activity in MuSCs with A-type lamin mutations responsible for severe congenital muscle dystrophy (L-CMD) in different conditions affecting the balance between nuclear import and export of YAP. Our study provides evidence that A-type lamin mutations impair YAP regulation by increasing the nuclear import of YAP. Intriguingly, we also found YAP nuclear accumulation in cells with nesprin-1 mutation responsible for a congenital myopathy and associated with defects in nuclear morphology [9,19], but not in cells carrying the *LMNA*^H222P^ mutation responsible for a less severe form of the disease and much milder nuclear envelope structural defects. These findings support a causative role of nuclear envelope defects in abnormal YAP signaling and implicated YAP as a pathogenic contributor in the severity of muscle dystrophies caused by nuclear envelop mutations. Overall, our study gains insight into broader questions of how lamins and nuclear shape impact cellular function.

## 2. Materials and Methods

### 2.1. Human Cells and Cell Culture

We obtained muscle biopsies from the Bank of Tissues for Research (Myobank, a partner in the EU network EuroBioBank) in accordance with European recommendations and French legislation. All patients provided written informed consent and experimental protocols were approved by our institution (INSERM) (approval number AC-2013-1868, 28 May 2014 and AC-2019-3502, 2 Dec 2019). Experiments were performed using immortalized L-CMD human myoblasts carrying a heterozygous *LMNA*c.94_96delAAG, p.Lys32del (referred to as ΔK32), *LMNA* p.Arg249Trp (referred to as R249W), or *LMNA* p.Leu380Ser (referred to as L380S) mutation. Immortalized human myoblasts carrying SYNE-1 homozygous c.23560 G<T, p.E7854X leading to a stop codon in exon 133 and deletion of the carboxy-terminal KASH domain (referred to as nesprin-1^Δ^^KASK^) were also analyzed, given that this mutation alters the nuclear shape of MuScs [9,20]. Immortalized myoblasts, obtained from two healthy control subjects without muscular disorders, were used as controls (hereafter referred to wild-type, WT).

We also analyzed myogenic cells derived from fibroblasts obtained from a patient with classical form of EDMD and carrying the *LMNA* p.H222P mutation (*LMNA*^H222P^), and from a control patient [21]. Fibroblasts were obtained from skin biopsies and immortalized as previously described [22]. Doxycycline-inducible Myod1 lentivirus was used to induce myogenic conversion [23].

Following muscular biopsy, MuSCs were immortalized and cultured in growth medium consisting of 1 vol 199 Medium /4 vol DMEM (Life Technologies, Carlsbad, CA, USA) supplemented with 20% fetal calf serum (Life Technologies), 5 ng/mL hEGF (Life Technologies), 0.5 ng/mL βFGF, 0.1 mg/mL Dexamethasone (Sigma-Aldrich, St. Louis, MO, USA), 50 µg/mL fetuin (Life Technologies), 5 µg/mL insulin (Life Technologies), and 50 mg/mL Gentamycin (GibcoTM, Life Technologies, Carlsbad, CA, USA). MyoD-transfected fibroblasts were cultured in a proliferation medium consisting of DMEM, supplemented with 10% fetal bovine serum (Life Technologies) and 0.1% gentamycin (Invitrogen, Carlsbad, CA, USA).

All cells were cultured on classic glass or plastic substrates. In addition, micro patterned glass slides with round islands of 700 µm^2^ (4D Cell, Montreuil, France) were coated with fibronectin and cells were seeded in a 200 µL drop at the center of the dish. After attachment, the wells containing the micro patterned slides were filled with proliferative medium for 24 h.

### 2.2. Drug Treatments

Importazole, a drug that blocks importin-β-dependent nuclear import [24], Leptomycin B, a drug that blocks CRM1-dependent nuclear export [25] or Dasatinib, Src-family kinase inhibitor drugs were diluted to final concentration of 40 µM, 100 nM, or 100 nM, respectively, for 24, 24, or 1 h. Latrunculin-A (Lat A, Sigma-Aldrich) or Cytochalasin D (Sigma-Aldrich) were diluted to final concentration of 2.0 and 1 µM, respectively, for 20 or 30 min. Vehicle control experiments using appropriate doses and time of dimethylsulfoxyde (DMSO) were used to assess the effects of specific drugs.

### 2.3. Luciferase Reporter Assays

MuSCs were transfected with Lipofectamine® 2000 (Invitrogen) reagents in growth media without antibiotics according to manufacturer’s instructions. TBS (Tead binding sequence: 14 times GGAATG)-Firefly Luciferase reporter constructs were used at a 1:5 ratio to the co-reporter vector for the weak constitutive expression of wild-type Renilla luciferase (pRL-TK, Promega GmbH, Mannhein, Germany). Transfected cells were seeded onto 24-, 48-, or 96-well plates and recovered overnight in growth medium. For the luciferase assay, cells were cultivated for 24 h after transfection under the stated conditions. The cells were lysed with passive lysis buffer (PJK GmbH, Kleinblittersdorf, Germany) and activity of the reporter was quantified by addition of firefly Luciferase substrate Beetle Juice (PJK GmbH). The activity of Renilla luciferase was quantified by addition of Renilla Juice (PJK GmbH) and measuring luciferase activity with Mithras LB940 Luminometer (Wildbad, Germany). Three separate experiments were performed per condition.

### 2.4. Immunocytochemistry and Image Analysis

MuSCs were fixed for 5 min with 4% formaldehyde, permeabilized with 0.1%Triton X100, and blocked with 10% bovine serum albumin (BSA) diluted in PBS. Cells were stained with Phalloidin-Alexa 568 to label F-actin (Interchim, Montluçon, France). The following primary antibodies were used for immunostaining: anti- Yes-associated protein (YAP)/transcriptional coactivator with PDZ-binding motif (TAZ) (Santa-Cruz, Dallas, TX, USA, sc-10119s), anti-phosphorylated Ser127 YAP (p^S127^-YAP) (Cell Signaling, Danvers, MA, USA, cs-4911), and anti-phosphorylated Tyr357-YAP (p^Y357^-YAP) (Abcam, Paris, France, ab62751). Secondary antibodies (Life Technologies, Saint-Aubin, France; 1/500) were Alexa Fluor 488 donkey anti-mouse IgG or Alexa Fluor 488 donkey anti-rabbit IgG. Nuclei were stained with Hoechst (ThermoFischer, Carlsbad, CA, USA) and Mowiol was used as mounting medium. Confocal images were taken with an Olympus FV 1200 (Olympus, Hamilton, Bermuda) and a laser-scanning microscopy Nikon Ti2 coupled to a Yokogawa CSU-W1 head (Nikon, Tokyo, Japan).

All image analyses were performed using Fiji software (NIH Image, version 1.51). For immunostained cells, Z-stacks of images were acquired for each channel, and the middle confocal slice was chosen from the images of the nucleus detected in the Hoechst channel. On the corresponding slice in the YAP channel, the average fluorescence intensity in the nucleus and just outside the nucleus (cytoplasm) was measured to determine the nuclear/cytoplasmic ratio. 

### 2.5. SDS-PAGE and Protein Analysis

Cells were lysed in total protein extraction buffer (50 mM Tris-HCl, pH 7.5, 2% SDS, 250 mM sucrose, 75 mM urea, 1 mM DTT) with added protease inhibitors (25 μg/mL Aprotinin, 10 μg/mL Leupeptin, 1 mM 4-[2-aminoethyl]-benzene sulfonylfluoride hydrochloride, and 2 mM Na_3_VO_4_) or directly in 2× Laemmli buffer. Protein lysates were separated by SDS-PAGE and transferred on PVDF or nitrocellulose membranes. After blocking with bovine serum albumin, membranes were incubated with anti-YAP (Santa-Cruz, CA, USA, sc-10119), anti-p^S127^-YAP (Cell Signaling, Danvers, MA, USA, cs-4911), and anti-p^Y357^-YAP (Abcam, Paris, France, ab62751) or anti-GAPDH (Cell Signaling, cs-2118). Goat anti-mouse, goat anti-rat, or donkey anti-goat HRP conjugates were used for HRP-based detection. Detection of adsorbed HRP-coupled secondary antibodies was performed by ECL reaction with Immobilon Western Chemiluminescent HRP Substrate (Millipore, Billerica, MA, USA). HRP signals were detected using a CCD-based detection system (Vilber Lourmat, Marne-La-Vallée, France) or a G-box system with GeneSnap software (Ozyme, Saint-Quentin, France). Membranes subjected to a second round of immunoblotting were stripped with stripping buffer (62.5 mM Tris-HCL pH 6.8, 2% sodium dodecyl sulfate (SDS), 100 mM β-mercaptoethanol) and incubated at 55 °C for 30 min with mild shaking before excessive washing with deionized water and re-blocking. Quantification was performed using ImageJ (NIH Image).

### 2.6. Quantification of Gene Expression

The mRNA was isolated from cell lysates using the RNeasy mini kit (Qiagen, Hilden, Germany) with the Proteinase K step, according to the manufacturer instructions. The complementary DNA (cDNA) was transcribed by SuperscriptIII (ThermoFischer, Carlsbad, CA, USA). Gene expression was quantified by using PerfeCTa-SYBR^®^Green SuperMix (Quanta, Biosciences, Gaithersburg, MD, USA) with the help of LightCycler 480 II (Roche Diagnostics GmbH, Mannheim, Germany). The primers were designed by Primer-BLAST (NCBI) and synthesized by Eurogentec (Liège, Belgium). Expression of all target genes was normalized to the expression of the reference gene RPLP0. Primer sequences are listed in Table 1.

### 2.7. Statistical Analysis

Graphpad Prism (Graphpad Software, La Jolla, CA, USA) was used to calculate and plot mean and standard error of the mean (SEM). Statistical significances were assessed by ANOVA followed by Bonferroni or two-tailed unpaired *t*-tests. Differences between conditions were considered significant at *p* < 0.05. Figures were plotted with Graphpad Prism.

## 3. Results

### 3.1. Impaired Yes-Associated Protein (YAP) Nuclear Exclusion in Confluent LMNA Mutant Muscle Stem Cells (MuSCs)

Wild-type (WT) MuSCs were plated either at low (10,000 cells/cm^2^) or high (40,000 cells/cm^2^) density and stained for YAP localization (Figure 1A,B). At low density, YAP was predominantly localized to the nucleus (Figure 1A,B). However, at high density conditions, WT cells showed predominantly cytoplasmic YAP, confirming previous reports for other cell types [26,27,28]. Similar YAP localization was observed in cells with the *LMNA*^H222P^ mutation (Appendix A) As expected, inhibition of CRM1-dependent nuclear export using Leptomycin B maintained preferential YAP nuclear distribution in confluent WT cells (Appendix A).

Interestingly, *LMNA* mutant cells plated at high density showed impaired density-dependent YAP subcellular localization and failed to exclude YAP from the nucleus (Figure 1A,B and Appendix A).

Lamin A/C’s are linked to the outer nuclear membrane protein nesprin-1 via SUN proteins in the lumen of the nuclear membrane. Nesprin-1 ^KASK^ mutation causes congenital myopathy and is also known to affect the nuclear shape [9,19]. Interestingly, nesprin-1 ^KASK^ cells displayed preferential nuclear YAP at high cell density (Figure 1B). Together, these finding revealed a striking correlation between the YAP mislocalization observed in vitro and the severity of the diseases.

Apart from cell–cell contacts, mechanical environments characterized by cell morphology and actin contractility regulate YAP nuclear localization [28]. Small cell surface adhesion is a known determinant for YAP nuclear exclusion [28]. Accordingly, WT cells on round micro-patterned surfaces of 700 µm^2^ displayed low nuclear staining of YAP (Figure 1C,D). In contrast, YAP was preferentially nuclear in *LMNA*^ΔK32^ cells cultured on small ECM substrates (Figure 1C,D). However, at low density, treatment with LatA induced YAP exclusion from the nucleus both in *LMNA*^ΔK32^ mutant and WT cells (Appendix A), thus supporting a dominant regulation of YAP localization by actin polymerization.

### 3.2. Phosphorylated Ser127-YAP Accumulates in the Nucleus of LMNA Mutated Muscle Stem Cells (MuSCs)

YAP phosphorylation on Ser 127 residue by LATS1/2 allows interaction with 14–3–3 protein and thereby nuclear exclusion of YAP [29]. We thus asked whether persistent nuclear localization in *LMNA*^ΔK32^ could be mediated by impaired LATS1/2 activity.

We found that under high density conditions *LMNA*^ΔK32^ cells accumulated p^S127^-YAP in the nucleus, in contrast to WT cells (Figure 2A,B). Interestingly, cell treatment with cytochalasin D a drug known to activate the Hippo pathway, increased the intensity of p^127^YAP staining (Appendix A), thus supporting the specificity of the p^S127^-YAP staining.

At low density, the amount of p^S127^-YAP did not differ between WT and *LMNA*^ΔK32^, and the level of p^S127^-YAP and the p^S127^-YAP/YAP ratio significantly increase with cell density in both cell lines (Figure 1C,D). This is in line with data reporting a cell density-dependent activation of the Hippo pathway [26].

### 3.3. Src-Dependent Tyr Phosphorylation of YAP is Activated in LMNA^ΔK32^ MuSCs

Aside from p^S127^-YAP, tyrosine phosphorylation of YAP by Src-kinase family modulates the transcriptional activity of YAP and indirectly its localization [30]. In both WT and *LMNA*^ΔK32^ cells, p^Y357^-YAP was predominantly localized to the nucleus in low density conditions (Figure 3A,B).

The nucleo-cytoplasmic ratio of p^Y357^-YAP significantly decreased at high densities in WT but not in *LMNA*^ΔK32^ mutant cells (Figure 3A,B). However, at the protein level, the p^Y357^-YAP/YAP ratio did not differ between cell lines and decreased with cell density in both cell lines (Figure 3C,D). Dasatinib, an Abl and Src-family kinase inhibitor, significantly reduced the amount of p^Y357^-YAP phosphorylation and the p^Y357^-YAP/YAP ratio (Figure 3F,G) as well as the nucleo-cytoplasmic ratio of p^Y357^-YAP (Figure 3A,H).

### 3.4. Blockade of Nuclear Import Inhibits Nuclear Accumulation of YAP in Confluent LMNA^ΔK32^ MuSCs

It is clearly established that YAP localization depends on its dynamic shuttling between the cytoplasm and the nucleus, and is maintained at a steady state by a balance between nuclear export and import rates. To exclude the possibility of nonspecific permeabilization of the nuclear envelope due to the lamin mutation, and generally to determine the exact pathway of nuclear import of YAP, we used importazole, an importin β-specific inhibitor [24]. In low density cultures, importazole inhibited YAP nuclear localization in both WT and *LMNA*^ΔK32^ cells, with no significant difference in YAP nucleo-cytoplasmic ratio between cell lines (Figure 4; each *p* < 0.001). In high density cultures, importazole significantly reduced nuclear localization of YAP in *LMNA*^ΔK32^ cells (Figure 4A,B, each
*p* < 0.001), but not in WT cells, thus attesting to a persistent nuclear import of YAP in high density *LMNA*^ΔK32^ cell cultures.

### 3.5. Functional Consequences of YAP Nuclear Accumulation in High Density LMNA^ΔK32^ MuSCs

To characterize the consequences of altered YAP translocation, we assessed expression of select YAP target genes and TEAD-dependent transcriptional activity using a luciferase reporter. Confluent *LMNA*^ΔK32^ cells had an increased expression of *YAP* and YAP target genes *CTGF* and *MYL9* (Figure 5A). Moreover, *LMNA*^ΔK32^ cells show increased TEAD-dependent luciferase reporter activity when compared to WT cells (Figure 5B), thus confirming elevated *YAP* transcriptional activity. Taken together, these data indicate that the *LMNA*^ΔK32^ mutation affects YAP localization and density-dependent inactivation of YAP.

## 4. Discussion

In this study, we showed that A-type lamin mutations which are responsible for congenital muscle disorders impacted the yes-associated protein (YAP) signaling pathway by increasing the nuclear import of YAP through the nuclear pore complexes. More importantly, YAP was transcriptionally active despite activation of the Hippo pathway, and thus may contribute to the impaired muscle differentiation in congenital muscle dystrophies. In addition, our data revealed a striking correlation between YAP deregulation, nuclear envelope defects, and disease severity, thus supporting a critical role of nuclear morphology in regulating YAP nuclear import.

### 4.1. Canonical Regulation of YAP Nucleo-Cytoplasmic Localization Via the Hippo Pathway

In skeletal muscle, YAP/transcriptional coactivator with PDZ-binding motif (TAZ) are powerful co-transcription factors which regulate muscle cell proliferation and differentiation [31,32] and play critical roles in controlling muscle growth [32,33]. The nuclear presence and the transcriptional activity of YAP/TAZ can be modulated by mechanical cues, such as substrate stiffness, cell spreading, and stretching (review in [34]) as well as by non-mechanical cues [26,35]. In previous work [10], we have shown that mutations in the *LMNA* gene associated to congenital muscle dystrophy cause a loss of environmental mechanosensing with elevated YAP signaling despite the soft environment, thus suggesting that A-type lamins modulate the mechanical regulation of YAP. Consistent with an abnormal mechanical regulation of YAP, we found that reducing cell spreading was ineffective to induce cytoplasmic relocalization of YAP in *LMNA*^ΔK32^ mutant cells (Figure 1C,D). In addition, here we also showed that cell–cell contact failed to inhibit YAP nuclear localization and activity in *LMNA* mutant cells from LMNA-related congenital muscular dystrophy (L-CMD), contrary to what was observed in our wild-type (WT) (Figure 1A,B) and *LMNA**^H222P^* (Appendix A) cells and in other non-cancer cells [26,27,28]. Taken as a whole, these data showed that defective nucleo-cytoplasmic shuttling of YAP was a hallmark of the most severe muscle dystrophies related to nuclear envelope mutations. Moreover, abnormal YAP regulation in *LMNA* mutant cells from L-CMD involved both mechanical and non-mechanical regulations of YAP nucleo-cytoplasmic shuttling. While our study mainly focused on YAP, whether mutations in nuclear envelope proteins also affect the nucleo-cytoplasmic shuttling of TAZ remain to be precisely determined.

The mechanisms by which cell density modulates YAP activation and nucleo/cytoskeletal shuttling have been extensively studied during the last decade [11,36,37]. The Hippo signaling pathway critically regulates cell–cell contact-mediated YAP cytoplasmic translocation [38,39,40]. In cells grown at low density, YAP is primarily localized to the nucleus where it promotes target gene transcription and proliferation. When cells reach a critical density, YAP translocates to the cytoplasm [26,27,28], thus underlying the classical paradigm of the contact inhibition of proliferation [26]. The Hippo signaling pathway functions as a highly conserved canonical upstream regulator of YAP activity and localization [41]. At the core of the Hippo pathway, LAT1/2 mediates serine phosphorylation on several serine residues, including serine 127, thus mediating YAP nuclear export and subsequent cytoplasmic association with 14–3–3 proteins [40,42,43,44]. In confluent cultures, loss of Ser127 phosphorylation and LATS1/2 activity with persistent nuclear localization of YAP is a hallmark of cancer cells [30]. In contrast, the high density cultures of *LMNA* mutated cells exhibited high protein levels of p^S127^-YAP and persistent nuclear localization of p^S127^-YAP (Figure 2), thus indicating activation of the Hippo pathway signaling. This nuclear accumulation of YAP and p^S127^-YAP in *LMNA*^ΔK32^ cells suggests that YAP nuclear export is insufficient to counterbalance YAP nuclear entry. Accordingly, it is known that the presence of p^S122^-YAP is a prerequisite, but is not sufficient for nuclear exclusion of YAP [45,46]. Although the role of exportin1 in YAP nuclear export has been clearly identified [29,47,48], however, regulation of YAP nuclear export remains largely unknown.

Regulation of YAP localization is also modulated by other kinases, including NLK [35] and Src-family kinases [47,49]. Whereas YAP phosphorylation on Ser residues are well-known negative regulators of YAP stability, Src-mediated phosphorylation of tyrosine 357 has been correlated with nuclear localization and increased YAP transcriptional activity [18,47,49]. Higher p^Y357^-YAP levels in *LMNA^DK32^* cells may thus contribute to the nuclear retention by promoting binding between YAP and TEAD transcription factors. Whether increased nuclear p^Y357^-YAP at both low and high cell densities could explain the increased transcriptional activity of YAP observed in *LMNA*^ΔK32^ cells (Figure 5) remains to be determined.

### 4.2. Nuclear Import of YAP in LMNA^ΔK32^ Mutant Cells

YAP nuclear import is mediated by active translocation involving importins through nuclear pores [50]. To interfere with active nuclear import, we used importazole, a drug known to inhibit importin-dependent nuclear translocation [24] and YAP nuclear localization [18]. In low density conditions, blocking nuclear entry considerably reduced nuclear localization of YAP in both WT and *LMNA* mutant cells (Figure 4). Thus, in low density conditions, YAP nuclear entry was mediated by an importin-dependent nuclear import in both WT and *LMNA*^ΔK32^ mutated MuSCs. Passive diffusion across the damaged nuclear envelope [51,52], if any, was not a main contributor of YAP nuclear import in *LMNA*^ΔK32^ mutant MuSCs. Moreover, importazole inhibited YAP nuclear localization in high density *LMNA*^ΔK32^ but not in WT cells. Therefore, in WT cells, at high cell density active nuclear import of YAP was inhibited and no further inhibition was observed after treatment with importazole. In contrast, active nuclear import of YAP persisted in *LMNA*^ΔK32^ cells, a finding consistent with a dominant active nuclear import of YAP at cell confluence (Appendix A). MuSCs carrying a mutation in the gene encoding the nuclear envelope protein nesprin1, also failed to regulate YAP localization, thus suggesting a nuclear envelope-related dysfunction.

Nuclear deformability is specific to *LMNA* and nesprin mutations rather than to muscular dystrophy and specifically affected the most severe forms of muscle disorders related to nuclear envelope defects [16,52]. Recent studies have reported that force-induced nuclear deformations increase YAP nuclear translocation through the nuclear pore complex [17,18]. It is thus conceivable that nuclear deformations per se drive nuclear translocation of YAP in *LMNA* and nesprin-1^ΔKASH^ mutant MuSCs (Appendix A), regardless of how nuclear deformation can be caused [53].

### 4.3. Functional Consequence of YAP Deregulation in Myogenesis

Nuclear localization of the transcriptional co-activator YAP and activation of the TEAD family transcription factors are required to promote proliferation but prevent differentiation of human stem cells [54,55]. In myogenic cell precursors cultured in vitro, high YAP expression and activity promotes proliferation of myogenic cell precursors whilst preventing their differentiation [31]. Therefore, one can speculate that persistent activation of YAP in *LMNA* mutant MuSCs has additive negative effects on skeletal muscle differentiation. Further studies are needed to precisely determine their contribution in the physiopathology of lamin-related muscle dystrophy.

## Figures and Tables

**Figure 1 cells-09-00816-f001:**
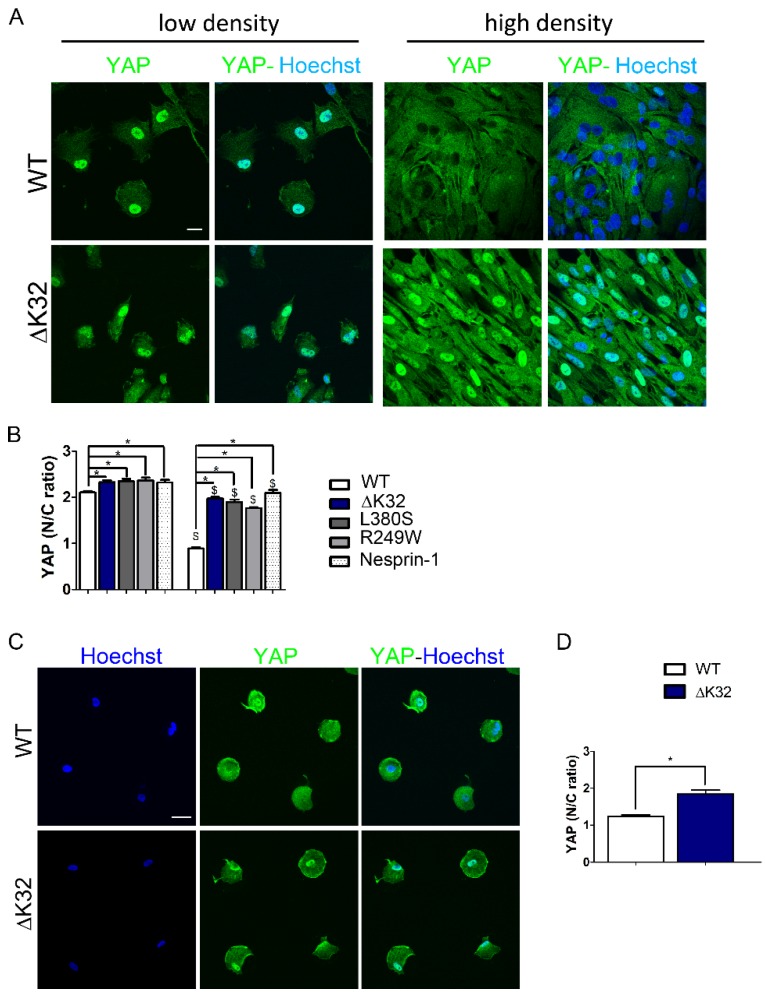
Modulation of yes-associated protein (YAP) localization in wild-type (WT) and mutant muscle stem cells (MuSCs). (**A**) Confocal images of YAP (green) in WT and *LMNA*^ΔK32^ mutant MuSCs cultured in low and high density conditions. Nuclei are stained with Hoechst (blue). Scale bar: 20 µm. (**B**) Quantification of YAP nucleo-cytoplasmic (N/C) ratio in WT, *LMNA*^ΔK32^, *LMNA*^L380S^, *LMNA*^R249W^, and nesprin-1 ^KASK^ mutant MuSCs. Pooled values of WT (WT1 and WT2) are presented. Values are expressed as mean ± SEM, *n* = 200 cells for each cell line. * *p* < 0.05 vs WT; $ < 0.05 vs. corresponding sparse condition. (**C**) Micro-patterning modulation of YAP. Confocal images of YAP (green) in WT and *LMNA*^ΔK32^ cells cultured on small ECM substrate that limits cell spreading. Nuclei are stained with Hoechst (blue). Scale bar: 30 µm. (**D**) Quantification of YAP nucleo-cytoplasmic (N/C) ratio on small ECM substrates. Values are expressed as mean ± SEM, *n* ≥ 62 cells for each cell line. * *p* < 0.05 vs. WT.

**Figure 2 cells-09-00816-f002:**
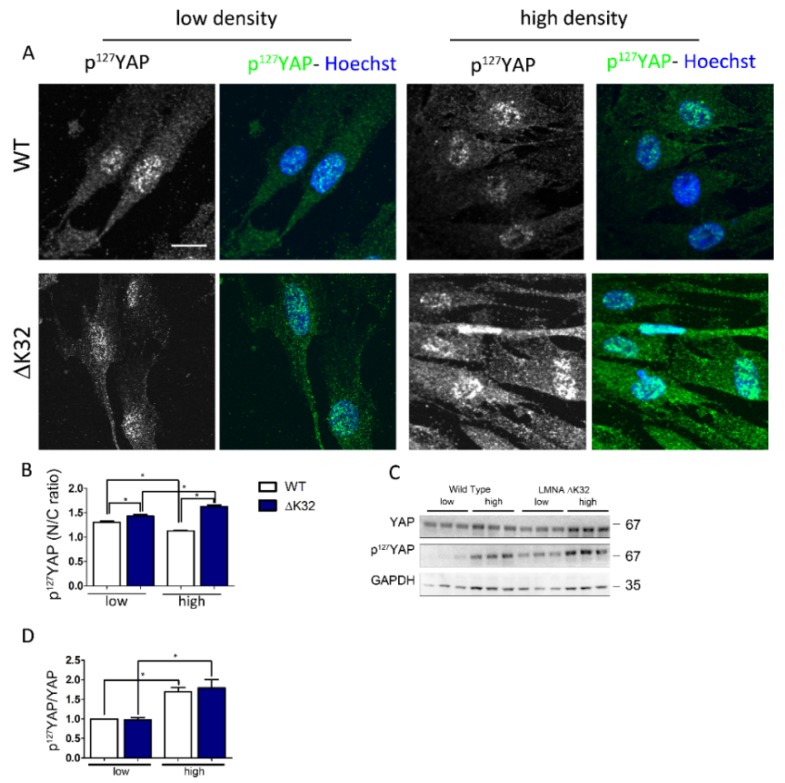
p^S127^-YAP accumulates in the nucleus of *LMNA*^ΔK32^ MuSCs. (**A**) Confocal images of p^S127^-YAP (green), in WT and *LMNA*^ΔK32^ mutant MuSCs cultured at low and high density conditions. Nuclei are stained with Hoechst (blue). Scale bar: 20 µm. (**B**) Quantification of p^S127^-YAP nucleo-cytoplasmic (N/C) ratio. Values are expressed as mean ± SEM, *n* = 150 cells for each cell line. * *p* < 0.005 compared with WT. (**C**) Representative Western-blot of YAP, p^S127^-YAP, and GAPDH in WT and *LMNA*^ΔK32^ MuSCs plated at low and high cell density. (**D**) Quantification of p^S127^-YAP /YAP protein levels in low and high density conditions, expressed in arbitrary units (a.u.). GAPDH was used as a loading control. Pooled values of WT (WT1 and WT2) are presented. Values are mean ± SEM, *n* ≥ 4 per conditions. * *p* < 0.005 compared with WT.

**Figure 3 cells-09-00816-f003:**
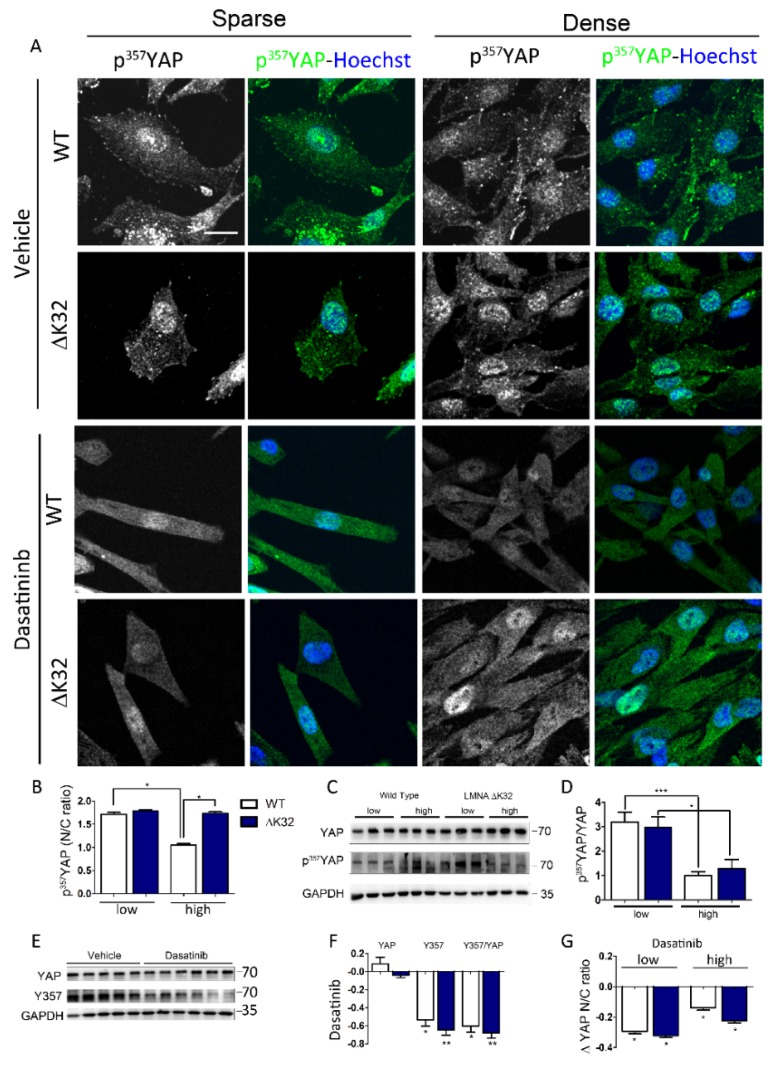
Y357 phosphorylation of YAP in WT and *LMNA* mutated MuSCs. (**A**) Confocal images of p^Y357^-YAP (green) in WT and *LMNA*^ΔK32^ mutant MuSCs cultured in low and high density conditions, at baseline and after treatment with Dasatinib. Nuclei are stained with Hoechst (blue). Scale bar: 20 µm. (**B**) Quantification of p^Y357^-YAP nucleo-cytoplasmic (N/C) ratio. Values are expressed as mean ± SEM, *n* = 150 cells for each cell line. (**C**) Representative Western-blot of YAP, p^Y357^-YAP, and GAPDH in WT and *LMNA*^ΔK32^ MuSCs plated at low and high cell density. (**D**) Quantification of p^Y357^-YAP/YAP protein levels in low and high density conditions, expressed in arbitrary units (a.u.). GAPDH was used as a loading control. Pooled values of WT (WT1 and WT2) are presented. Values are mean ± SEM, *n* ≥ 4 per conditions. * *p* < 0.005, ** *p* < 0.01, *** *p* < 0.001 compared with WT. (**E**) Representative Western-blot of YAP, p^Y357^-YAP, and GAPDH in WT and *LMNA*^ΔK32^ MuSCs treated with vehicle or Dasatinib. (**F**) Quantification of YAP, p^Y357^-YAP, and p^Y357^-YAP /YAP protein levels after treatment with Dasatinib. Values are expressed as percent change of values obtained without treatment. GAPDH was used as a loading control. Pooled values of WT (WT1 and WT2) are presented. Values are mean ± SEM, *n* ≥ 4 per conditions. * *p* < 0.005, ** *p* < 0.01, compared with WT. (**G**) Quantification of p^Y357^-YAP nucleo-cytoplasmic (N/C) ratio after Dasatinib treatment expressed as a fraction of control value obtained in sparse or dense conditions before treatment. Values are expressed as mean ± SEM, *n* ≥ 110 cells for each cell line.

**Figure 4 cells-09-00816-f004:**
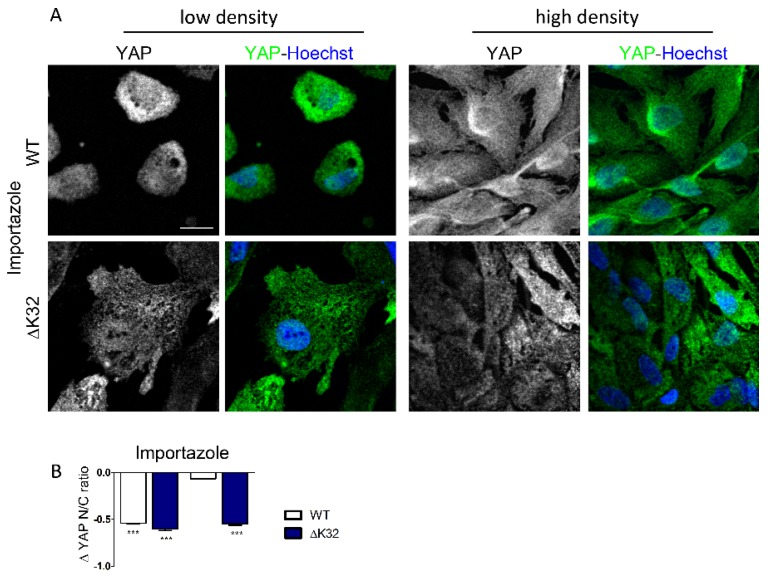
Importazole inhibits nuclear localization of YAP in WT and mutant MuSCs. (**A**) Confocal images of YAP (green) in WT and *LMNA*^ΔK32^ mutant MuSCs cultured in low and high density conditions. Nuclei are stained with Hoechst (blue). Scale bar: 20 µm. (**B**) Quantification of YAP nucleo-cytoplasmic (N/C) ratio after importazole treatment. Values are expressed as mean ± SEM, as a fraction of value obtained before importazole treatment, *n* ≥ 60 cell in each cell line.

**Figure 5 cells-09-00816-f005:**
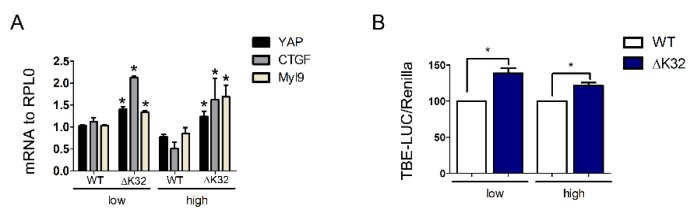
Transcriptional activity of YAP. (**A**) Histogram represents mRNA transcripts of *YAP*, *CTGF*, and *Myl9* normalized to *RPLP0* and expressed as fold-changes. Values are ± SEM, *n* = 3 separate experiments. * *p* < 0.05 compared with WT. (**B**) Quantification of *YAP* activity using the TBE-Luciferase (LUC)/Renilla reporter. Values are ± SEM, *n* = 3 separate experiments. * *p* < 0.05 compared with WT cells.

**Table 1 cells-09-00816-t001:** Primer sequences.

Gene name	Abbreviation	Forward/Reverse	Sequence
Human Yes-associated protein 1	hYAP1	fw	GCTACAGTGTCCCTCGAACC
		rev	CCGGTGCATGTGTCTCCTTA
Human Connective tissue growth factor	h-CTGF	fw	ACCGACTGGAAGACACGTTTG
		rev	CCAGGTCAGCTTCGCAAGG
Human Connective tissue growth factor	h-RPLPO	fw	CTCCAAGCAGATGCAGCAGA
		rev	ATAGCCTTGCGCATCATGGT
Human Glyceraldehyde 3-phosphate dehydrogenase	h-GAPDH	fw	TGC-CAT-GTA-GAC-CCC-TTG-AA
		rev	TGG-TTG-AGC-ACA-GGG-TAG-TT
Human Myosin light chain 9	h-Myl9	fw	CGA-ATA-CCT-GGA-GGG-CAT-GAT
		rev	AAA-CCT-GAG-GCT-TCC-TCG-TC

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
