# Peer review of "Lamin Mutations Cause Increased YAP Nuclear Entry in Muscle Stem Cells"

_cells, 2020, doi:10.3390/cells9040816_

Round 1

Reviewer 1 Report

The manuscript entitled ‘Lamin Mutations Cause Increased YAP Nuclear Entry in Muscle Stem Cells’ by Owens D.J. et al. represents an interesting manuscript where the authors analysed different conditions affecting the balance between nuclear import and export of YAP. However, I considered these studies interesting, they should become more consistent and robust.

Major concerns:

  1. Figure 1A, the images presented for WT and ΔK32 have clearly different cell densities.
  2. Regarding the cell counts: How many different experiments were performed? Only 60 cells were analysed/counted in Figure 1B and 35 cells in Figure 1D? These numbers are clearly very low.
  3. In figure 1C only one cell is presented! And back and white images mixed with coloured ones are not a good option.
  4. The figure 2 and figure 3 are very difficult to follow.
  5. In Figure 3A the immunostaining for pS127 in the ΔK32 are not very good. Have the authors tested another antibody?
  6. The immunoblots are very little and so difficult to interpret. Regarding the immunoblots quantification only the ratio should be presented. The D and E should be removed. The pS127 YAP levels are determined classically determined by the ratio between the phosphorylated form and the total protein levels.
  7. The ICQ results are not corroborated by the WB results.Could the authors explain?
  8. In figure 2 the number of experiments performed is not indicated, and the number of cells counted are very low.
  9. In figure 2, the levels of YAP phosphorylation are increased in both WT and ΔK32 at high density conditions. The authors should explain this result.
  10. The figure 3 should be reorganized as previously suggested to Figure 2., for example.

Minor concerns:

  1. Means+ SEM change to mean +SEM

Author Response

We thank Reviewer 1 for their helpful comments, all of which have been taken into consideration in the answers as well as in the revised version of the manuscript.

Major concerns:

  1. Figure 1A, the images presented for WT and ΔK32 have clearly different cell densities.

R : Fig. 1A has been modified. In the revised version, images presented for WT and DK32 have similar cell densities.

  1. Regarding the cell counts: How many different experiments were performed? Only 60 cells were analysed/counted in Figure 1B and 35 cells in Figure 1D? These numbers are clearly very low.

R: The authors thank the reviewer for this suggestion. All experiments were performed at least 3 times. Additional cells have now been analyzed (n=200 for each cell line in Fig. 1B and n=60 in Figure 1D).

In figure 1C only one cell is presented! And back and white images mixed with coloured ones are not a good option.

R: Thank you for this suggestion to improve the figure. The figure has now been modified accordingly.

  1. The figure 2 and figure 3 are very difficult to follow.

R: Figures 2 and 3 have been modified as recommended by the Reviewer (please see below).

In Figure 3A the immunostaining for pS127 in the ΔK32 are not very good. Have the authors tested another antibody?

We tested different antibodies anti- p127YAP antibody including ab138672 (Abcam) and cs-4911 (Cell signaling). The choice of anti- p127YAP from Cell Signaling was based on preliminary data showing that treatment with cytochalasin D, a drug known to activate the Hippo pathway increased the intensity of p127YAP staining, as expected. These additional experiments have been added in suppl Fig. 3

In addition, using this antibody we previously showed that reducing YAP expression by using siRNA against YAP reduced the YAP staining in muscle cell precursors (Bertrand et al, 2014).

The immunoblots are very little and so difficult to interpret. Regarding the immunoblots quantification only the ratio should be presented. The D and E should be removed. The pS127 YAP levels are determined classically determined by the ratio between the phosphorylated form and the total protein levels.

The authors agree with the reviewer and the Figure has been modified as recommended.

  1. The ICQ results are not corroborated by the WB results.Could the authors explain?

The authors thank the Reviewer for their helpful comments and apologize for the mistake. We have now carefully rechecked all of the data. In the revised version, ICQ results are corroborated by the WB results. 

  1. In figure 2 the number of experiments performed is not indicated, and the number of cells counted are very low.

The authors recognize this oversight and the number of experiments performed is now indicated in the revised manuscript (n=3 independent experiments). In addition, additional cells were analyzed (final n=150 for each cell line).

  1. In figure 2, the levels of YAP phosphorylation are increased in both WT and ΔK32 at high density conditions. The authors should explain this result.

The pS127-YAP/YAP ratio increased significantly with cell density in both cell lines, in agreement with a cell-dependent activation of the Hippo pathway. However, in high cell density conditions LMNAΔK32 cells accumulated pS127-YAP in the nucleus, in contrast to WT cells. This indicates that LMNADK32 mutations failed to exclude YAP from the nucleus at high cell density, despite activation of the Hippo pathway.

The figure 3 should be reorganized as previously suggested to Figure 2., for example.

The authors agree with the reviewer and Figure 3 has been reorganized as suggested.

Minor concerns:

  1. Means+ SEM change to mean +SEM

Thank you, this has been rectified.

Reviewer 2 Report

This manuscript by Owens et. al. describes that at least some LMNA mutations increase nuclear import of the transcriptional regulator YAP leading to more nuclear YAP under conditions that should promote cytoplasmic relocalization. These effects are reported to occur in an actin cytoskeleton-dependent manner. Immortalized human myocytes from patients or myocyte-like cells that were derived from patient fibroblasts were utilized for these studies. Overall, there are some concerns with the presentation of the data and more importantly the limited extent of novel mechanistic insights provided by these studies that temper my enthusiasm for the manuscript.

Specific comments:

Could the authors validate their antibodies with some kind of knockdown experiments since all of the studies are based on their use.

For all of the YAP N/C quantifications I am confused by the magnitude of the changes. Based on the representative images provided, in fig 1A for the WT cells there is a profound nuclear localization and I would expect more than a N/C ratio of 2. And at high confluency there is a profound loss of nuclear signal and I would expect a ration to be far less than 0, let alone the provided ratio of 1 (a ratio of 1 would have the nucleus and cytoplasm looking similar in their fluorescence intensity and there is a clear lack of nuclear labeling.

Can images for S2C (latrunculin experiments) be shown? The change might be significant but it is quite subtle. I would be reuctant to make any statement about the role of actin based on this subtle change.

On page 7 there is reference to data from fig 2 that is written as being found in figure 1. Similar issue on page 9 where reference to fig 3 data is called fig 5.

That Yap nuclear localization is importin dependent is not surprising as the protein is large enough to likely require active transport through the NPC.

For fig 5A there are no indications of statistical significance shown on the graph itself.

Author Response

The authors thank Reviewer 2 for helpful comments, all of which have been taken into account in the answers as well as in the revised version.

Specific comments:

  1. Could the authors validate their antibodies with some kind of knockdown experiments since all of the studies are based on their use.

Validation of the antibodies was based on different sets of experiments:

  1. We compared YAP staining after siRNA against YAP or control siRNA against LacZ(Bertrand et al, 2014) in MuSCs: Reducing total YAP reduced YAP staining in MuSCs.
  2. Increased p127YAP through activation of the Hippo pathway: Cells were also treated for 10 min with cytochalasin D, a drug known to activate the Hippo pathway. As expected, we found an increased p127YAP staining after Cytochalasin treatment.  These data have been added in the Rvised manuscript (Figure S3).
  • More importantly, the use of p127YAP for IF has already been published (Ege et al 2018).
  1. Similar results were obtained using two different anti-YAP antibodies (D8H1X XP® Ozyme and Santa Cruz Biotechnology, Santa Cruz, CA, USA).
  2. For all of the YAP N/C quantifications I am confused by the magnitude of the changes. Based on the representative images provided, in fig 1A for the WT cells there is a profound nuclear localization and I would expect more than a N/C ratio of 2. And at high confluency there is a profound loss of nuclear signal and I would expect a ration to be far less than 0, let alone the provided ratio of 1 (a ratio of 1 would have the nucleus and cytoplasm looking similar in their fluorescence intensity and there is a clear lack of nuclear labeling.

The authors thank the reviewer for identifying areas that need clarification. Additional cells have been analyzed (n=200 for each cell lines in Fig. 1B and n=60 in Figure 1D). We found a N/C ratio close to 2 in low density conditions, as indicated by the Reviewer. At high density (confluency), there is a profound loss of the nuclear signal in a subset of WT cells, but there is also a subset of cells with nucleus and cytoplasm with a similar intensity of fluorescence (Fig. 1A), consequently the mean±SE N/C ratio in confluent cells was 0.89±0.02. WT cells on round micro-patterned surfaces of 700 µm2 µm2 displayed similar intensity of nuclear and cytoplasmic fluorescence (Fig. 1C).

  1. Can images for S2C (latrunculin experiments) be shown? The change might be significant but it is quite subtle. I would be reuctant to make any statement about the role of actin based on this subtle change.

Images obtained after latrunculin A treatment have been added in the revised version, as recommended.

  1. On page 7 there is reference to data from fig 2 that is written as being found in figure 1. Similar issue on page 9 where reference to fig 3 data is called fig 5.

Typo errors have been corrected.

  1. That Yap nuclear localization is importin dependent is not surprising as the protein is large enough to likely require active transport through the NPC.

The authors agree with this comment. However, given that the lamin mutation can be associated with nuclear envelope defects, it was important to validate that YAP active transport occurs through the NPC also in mutant cells

  1. For fig 5A there are no indications of statistical significance shown on the graph itself.

Statistical significance has been added, as recommended.

Reviewer 3 Report

This manuscript describes the intrinsic mechanisms by which lamins influence YAP subcellular distribution, by analyzing different conditions affecting the balance between nuclear import and export of YAP. The authors have previously reported that the mutations in the LMNA gene associated to congenital muscle dystrophy cause a loss of environmental mechanosensing with elevated YAP signaling despite soft environment, thus suggesting that A-type lamins modulate the mechanical regulation of YAP. In this manuscript, the authors showed that A-type lamin mutations which are responsible for congenital muscle disorders impacted the YAP signaling pathway by increasing the nuclear import of YAP through the nuclear pore complexes. The findings in this manuscript are important to understand the mechanisms of severe skeletal muscle laminopathies and would contribute to develop the treatment for the disease.

There are some minor comments to be considered before accepting the manuscript.

  1. There are some garbling in the notation indicating the mutant type, such as nesprin-1ΔKASH (Line 25), ΔK32 (Line 85).
  2. Line 90, nesprin-1ΔKASH instead of nesprin-1ΔKASK?
  3. It should be specified that “L-CMD mutation” is LMNAΔK32 in the text. In addition, it would be better to describe the relationship between the mutation, including L380S and R249W, and the symptom precisely in Introduction or Results.
  4. What is the abbreviation for TBE assay (Line 123)?
  5. Lat A appears on Line 208, but there is no description of what it is and what it does.
  6. The explanations of Figs. 3G and H are insufficient and difficult to understand.
  7. “Fig. 2” on Line 273 seems to be “Fig.4”. There were some more similar mistakes.
  8. The explanation of Fig. 4B is insufficient and difficult to understand. In addition, there is no explanation for which each column means.
  9. The description on Line 345, “Src inhibitors reduced the nuclear localization of YAP in both WT and LMNA cells”, is correct for low density in Fig. 3H, but seems to be opposite for high density.

Author Response

We thank Reviewer 3 for helpful comments, all of which have been taken into account in the answers as well as in the revised version.

  1. There are some garbling in the notation indicating the mutant type, such as nesprin-1ΔKASH (Line 25), ΔK32 (Line 85).

This has been modified.

  1. Line 90, nesprin-1ΔKASH instead of Nesprin-1ΔKASK?

This has been modified.

  1. It should be specified that “L-CMD mutation” is LMNAΔK32 in the text. In addition, it would be better to describe the relationship between the mutation, including L380S and R249W, and the symptom precisely in Introduction or Results.

 Specification has been done throughout the text.

  1. What is the abbreviation for TBE assay (Line 123)?

TBE stands for Tris-borate-EDTA, the buffer used for the luciferase experiment. This has been modified.

  1. Lat A appears on Line 208, but there is no description of what it is and what it does.

This has been corrected (line 119)

  1. The explanations of Figs. 3G and H are insufficient and difficult to understand.

Explanations have been added in the revised version. We hope this clarifies.

  1. “Fig. 2” on Line 273 seems to be “Fig.4”. There were some more similar mistakes.

Figure citations have been carefully checked

  1. The explanation of Fig. 4B is insufficient and difficult to understand. In addition, there is no explanation for which each column means.

Explanations have been added in the revised version. We hope this clarifies.

  1. The description on Line 345, “Src inhibitors reduced the nuclear localization of YAP in both WT and LMNA cells”, is correct for low density in Fig. 3H, but seems to be opposite for high density.

We agree that Src inhibitors reduced the nuclear localization of YAP in both WT and LMNA cells at low densities (line 355). This has been modified.

Reviewer 4 Report

The manuscript by Daniel Owens and colleagues from the laboratory of Catherine Coirault describes the effect of mutated Lamin and Nesprin, two nuclear envelope proteins, on the regulation of YAP, one of the two effectors of the Hippo tumor suppressor pathway. Using muscle stems cells as well as cells derived from patients with severe congenital muscle dystrophy (with mutations within the Lamin and Nesprin genes) and other well-engineered cell models, that authors showed that the mutant-harboring cells lose the subcellular regulation of YAP protein localization, compared to control cells.  An overarching hypothesis is that certain mutants of Lamin and Nesprin that compromise proper formation of nuclear envelope deregulate mechano-sensing properties of the Hippo signaling pathway.  

The manuscript is well presented and the quality of the data is fine.  Even though the actual mechanism, by which the mutated Lamin and Nesprin cause the abnormal localization of YAP in response to the cell density and/or surface of the cell spreading is not clear, the phenomenon uncovered by this study is interesting and will lead to the better description of molecular pathology behind certain forms of muscular dystrophy.

The following minor changes are suggested to improve the manuscript:

  1. It is important to show molecular weight markers in the gel fragments shown in several figures.  These markers could be generated by interpolation of the actual markers.
  2. A simple figure at the end with the model proposed by this work would make this report stronger.  (A sort of visual “abstract” of the work at the end of the discussion).
  3. If space allows, it would be important to discuss the functional mapping of actual Nuclear Import and Nuclear Export Sequences (NIS & NES) of YAP as published by M Kofler from A Kapus lab (Nature Communications 2018) in terms how could play a role, if at all, in the observed findings.
  4. Also, the presence of Serine 127 in YAP which is phosphorylated by Nemo-Like Kinase and could override or act independently from p-Ser128 signal, deserves to be discussed and considered in future experiments (S Moon from Jho lab - EMBO Reports, 2017 18(1) 61-71.
  5. Again if space allows a relevant review on “The Hippo signal transduction network in skeletal and cardiac muscle”
 by H Wackerhage and colleagues from J Sadoshima lab (Science Signalling 2014 337) could be mentioned.
  6.  What about TAZ, a paralogue of YAP, which was shown to respond to mechanical cues as well as YAP. A short discussion of TAZ would be appropriate.   

Author Response

We thank Reviewer 4 for helpful comments, all of which have been taken into account in the answers as well as in the revised version.

  1. It is important to show molecular weight markers in the gel fragments shown in several figures.  These markers could be generated by interpolation of the actual markers.

Thank you for this suggestion. Molecular weight markers have been added, as recommended.

  1. A simple figure at the end with the model proposed by this work would make this report stronger.  (A sort of visual “abstract” of the work at the end of the discussion).

The authors thank the reviewer for this suggestion and agree that this makes a valuable addition to the manuscript. As such a visual abstract has now been created (Fig. S4)

  1. If space allows, it would be important to discuss the functional mapping of actual Nuclear Import and Nuclear Export Sequences (NIS & NES) of YAP as published by M Kofler from A Kapus lab (Nature Communications 2018) in terms how could play a role, if at all, in the observed findings.

The authors thank the reviewer for this suggestion. Please note that the work by Kofler et al was already cited in the manuscript. Due to space limitation, it is difficult to further develop their results. Discussion about their results may play a role in our observed findings is highly speculative yet. 

  1. Also, the presence of Serine 127 in YAP which is phosphorylated by Nemo-Like Kinase and could override or act independently from p-Ser128 signal, deserves to be discussed and considered in future experiments (S Moon from Jho lab - EMBO Reports, 2017 18(1) 61-71.

The authors thank the reviewer for this suggestion. Reference has been added, as recommended. As stated before, due to space limitation, it is difficult to further develop their results and interpretation in the context of our findings.

  1. Again if space allows a relevant review on “The Hippo signal transduction network in skeletal and cardiac muscle” by H Wackerhage and colleagues from J Sadoshima lab (Science Signalling 2014 337) could be mentioned.

This has been done (line 417).

  1.  What about TAZ, a paralogue of YAP, which was shown to respond to mechanical cues as well as YAP. A short discussion of TAZ would be appropriate.   

The following sentence has been added in the revised manuscript:  While our study mainly focused on YAP, whether mutations in nuclear envelope proteins also affect the nucleo-cytoplasmic shuttling of TAZ remain to be precisely determined (lines 431-433).

Round 2

Reviewer 1 Report

Please find my comments about the revision of the manuscript entitled Lamin Mutations Cause Increased YAP Nuclear Entry in Muscle Stem Cells’ by Owens D. J. et al. The authors respond to all issues raised in my revision and performed the adequate alterations of the manuscript and it is now significantly improved.

Just a minor point they should include the number of cells counted in each experiment in the figure legend.

Overall, I believe that the manuscript is ready for publication.

Author Response

The authors thank Reviewer 1 for his/her helpful comment. The number of cells counted in each experiment has been added, as recommended.

Reviewer 2 Report

The authors have addressed the critiques from my initial review. I am still surprised at the quantification outcomes for various IF and IB experiments reported in many of the figures as compared to the representative images, but suspect that these don't change the fundamental results.

Author Response

The authors thank Reviewer 2 for his/her helpful comments and careful reading of our manuscript.